# Spontaneous Resolution of Late-Onset, Symptomatic Fluid Collection Localized in the Meningioma Resection Cavity: A Case Report and Suggestion of Possible Pathogenesis

**DOI:** 10.3390/brainsci11030299

**Published:** 2021-02-27

**Authors:** Yeong Jin Kim, Tae-Young Jung, In-Young Kim, Shin Jung, Kyung-Sub Moon

**Affiliations:** Department of Neurosurgery, Chonnam National University Research Institute of Medical Science, Chonnam National University Hwasun Hospital and Medical School, Hwasun 58128, Korea; mbdosa88@naver.com (Y.J.K.); jung-ty@jnu.ac.kr (T.-Y.J.); kiy87@hanmail.net (I.-Y.K.); sjung@jnu.ac.kr (S.J.)

**Keywords:** conservative management, convexity meningioma, subdural fluid collection, postoperative complication

## Abstract

Postoperative complications after brain tumor surgery occur occasionally and it is important for clinicians to know how to properly manage each complication. Here, we described a rare case of late-onset, subdural fluid collection localized at the resection cavity that caused motor weakness after convexity meningioma resection, requiring differentiation from an abscess, to help clinicians determine treatment strategies. A 58-year-old right-handed female was admitted to the hospital with a headache and posterior neck pain. Brain computed tomography (CT) scans and magnetic resonance (MR) images showed a homogeneously enhanced, calcified, and multi-lobulated mass adjacent to the right motor strip without perilesional edema. The patient underwent surgery without incident or residual deficit and was discharged from the hospital in good condition. Six weeks after surgery, the patient complained of left arm monoparesis without infection-related symptoms. Brain imaging studies showed a localized fluid collection in the resection cavity with an enhanced margin and perilesional edema. Diffusion restriction was not detected. After three months of conservative treatment without surgery or antibiotics, she recovered from the neurologic deficits, and brain imaging studies showed the spontaneous regression of the fluid collection and perilesional edema. Late-onset, localized fluid collection at the resection cavity, which is similar to an abscess, could occur three to eight weeks after meningioma resection. When there are predisposing factors, including blood components and hemostatic materials in the surgical cavity, it is important for clinicians to understand this type of complication and choose conservative management as a feasible strategy.

## 1. Introduction

After craniotomy for brain tumor removal, postoperative complications occasionally occur. For appropriate management, the clinician should be well-aware of the types, differential diagnosis, treatment strategies, and course of the complications [1]. It is especially important to distinguish minor complications, treated by conservative management, from major complications, which should be treated surgically. Reoperation is a huge burden for both the operator and the patient. Symptomatic and localized fluid collection at the resection site is a rare complication in intracranial meningioma surgery, which is different from major complications, such as abscesses, hemorrhages, or venous infarctions, and can be treated by conservative management. We described a rare case of late-onset, localized subdural fluid collection in the resection cavity causing motor weakness six weeks after convexity meningioma resection. The aim of this study was to remind clinicians that this kind of complication could be treated with a conservative strategy, not with a surgical approach.

## 2. Case Presentation

A 58-year-old right-handed female presented to the clinic with a headache of long duration and a two-week history of posterior neck pain. She had no significant medical or family history. No other neurological deficits were noted. Brain computed tomography (CT) scans and magnetic resonance (MR) images demonstrated a homogeneously enhanced, calcified, and multi-lobulated mass adjacent to the right motor strip without peritumoral edema (Figure 1A–C). The patient underwent a craniotomy for tumor resection without incident or residual deficit. The arachnoid plane between the multi-lobulated tumor and the motor strip was well-preserved. The surrounding vascular structures were also not injured during surgery except for venous bleeding from the mass stump, which was controlled by a fibrin sealant patch and oxidized cellulose (Figure 1D). A dura mater defect was repaired with an artificial dura substitute. There were no abnormal changes on intraoperative monitoring for motor-evoked potential and somatosensory-evoked potential during surgery. The postoperative CT scans showed the usual postoperative changes without hemorrhage or infarction (Figure 1E). Histopathologically, the mass was confirmed as fibrous meningioma. The patient recovered well uneventfully.

Six weeks after surgery, the patient complained of sudden left arm monoparesis. The CT scans and MR images revealed localized subdural fluid collection at the resection cavity with enhanced resection margin and perilesional edema, showing an iso-signal intensity of the cerebrospinal fluid (CSF) on T2-weighted and T1-weighted images. Diffusion restriction was not noticed (Figure 2A–C). There were no abnormal laboratory findings in the blood samples. The surgical wound was clean and well healed. After two weeks of conservative treatment with mannitol and steroids, the motor symptoms were completely recovered. The CT scans demonstrated decreased resection cavity size and marginal enhancement, but perilesional edema still existed (Figure 2D). Two weeks after the CT scans, the patient unfortunately presented with recurring left arm monoparesis and tingling sensations. The CT scans showed an increased amount of fluid in the cavity with perilesional edema (Figure 2E). After another two months of conservative treatment with prednisolone at 10 mg per day, she again fully recovered from the neurologic deficits. The CT scans showed decreased resection cavity size and perilesional edema (Figure 2F). Since then, the patient has had no specific neurologic deficits without specific medication. The latest MR images taken one year after operation revealed the complete disappearance of the fluid collection and perilesional edema with no recurrence (Figure 2G–I). 

## 3. Discussion

Complications after craniotomy surgery are not rare and are well-known to clinicians. The resection of convexity meningioma can induce several well-known site-specific complications, including abscesses, hemorrhages, or venous infarctions [2]. We presented a patient with late-onset and symptomatic fluid collection, localized at the resection site after convexity meningioma surgery. Because this type of complication has rarely been reported to date, our report might contribute to the decision-making for treating patients with this unusual complication. 

The underlying mechanisms of this complication are suspected to include many causative factors. Like subdural hygromas and tumor bed cysts, arachnoid tears and flap valves might be triggering factors. Sufficient dura-arachnoid interspace after the removal of a convexity meningioma might contribute to subdural fluid collection [3,4]. Subdural hygromas and tumor bed cysts have different clinical features and need different treatment approaches. Other factors also affect late-onset and localized fluid collection at the resection site. 

After meningioma surgery, the surgical site shows a well-enhanced, thickened meningeal membrane by three to eight weeks after surgery, which usually resolves after six months or more. Usually, well-enhanced granulation tissue is replaced by less-enhanced collagen. Furthermore, when there are blood components in the subdural fluid, persistent membrane enhancement is more prominent. This strong enhancement of the meninges after surgery is due to reactive granulomatous inflammation limited to the surgical site [2]. Reactive granulomatous inflammation is reinforced by hemostatic agents and artificial dura. Local inflammation is more likely to occur when many different types of exogenous materials are used. Additionally, a local allergic response to exogenous materials could be caused by individual-specific factors [5]. In our case, bleeding from the mass stump was controlled with a fibrin sealant patch and oxidized cellulose, and the dura defect was substituted with artificial dura material. An arachnoid tear might have been induced at the bleeding site with flap valve formation. Brain CT scans immediately after the tumor removal showed surgical site subdural fluid collection. Collectively, these multi-factors contributed to symptomatic fluid collection localized at the resection cavity in meningioma surgery in our case (Figure 3). 

It is difficult to distinguish imaging features of persistent subdural fluid collection due to granulomatous inflammation, subdural empyema, or tumor recurrence. Brain MR imaging techniques, including T1- or T2-weighted, fluid-attenuated inversion recovery (FLAIR), and diffusion weighted images (DWIs), are helpful in distinguishing empyema or tumor recurrence, but this is not always the case [6]. Brain imaging, blood laboratory test results, and the clinical course should be considered in the differential diagnosis. In this case, the blood infection markers, C-reactive protein (CRP), erythrocyte sedimentation rate (ESR), and leukocyte count, were within the normal range and the skin wound site was clear without redness, heat, tenderness, or swelling. The progression of fibrous meningioma within two months is very rare. With the lack of diffusion restriction and intact dura on brain imaging, the clinical course and laboratory tests suggested that subdural fluid collection was less likely to be empyema or a hemorrhagic complication.

Because of deficient examples in the pertinent literature, definitive treatment strategies for this complication are not decided yet. Symptomatic necrotizing lesions caused by granulomatous inflammation can be treated with surgical intervention to exclude brain abscesses [7]. As shown in usual cases, however, granulomatous inflammation spontaneously resolves and is replaced by collagen with time [2]. With steroid administration, inflammation could regress more dramatically. After the regression of inflammation, the subdural fluid could decrease by re-expansion of the brain and absorption of the subdural fluid. As a thickened meningeal membrane naturally resolves with time, the adjacent brain might re-expand with the absorption of subdural fluid. Sometimes, subdural hygroma is refractory to conservative treatment and surgical management is required [8]. Relatively, early-onset subdural cystic fluid collection grows rapidly, inducing sudden neurological symptoms, and is treated by surgical approaches [9]. Late-onset localized fluid collection at the resection cavity, which responds well to medical management, might resolve on its own, and a conservative approach appears to be a feasible strategy as in our case. Close clinical observation is suggested until the regression of the meningeal enhancement and perilesional edema induced by the inflammatory reaction. It is vital for clinicians to be aware of this complication and consider conservative management as a feasible strategy.

## 4. Conclusions

In conclusion, late-onset subdural fluid collection localized in the resection cavity, similar to an abscess, can occur after convexity meningioma surgery. Blood components and exogenous materials in the resection cavity may exacerbate CSF accumulation and induce neurologic symptoms. Even though neurologic symptoms are triggered by this complication, it is important for clinicians to choose conservative management as an initial treatment strategy.

## Figures and Tables

**Figure 1 brainsci-11-00299-f001:**
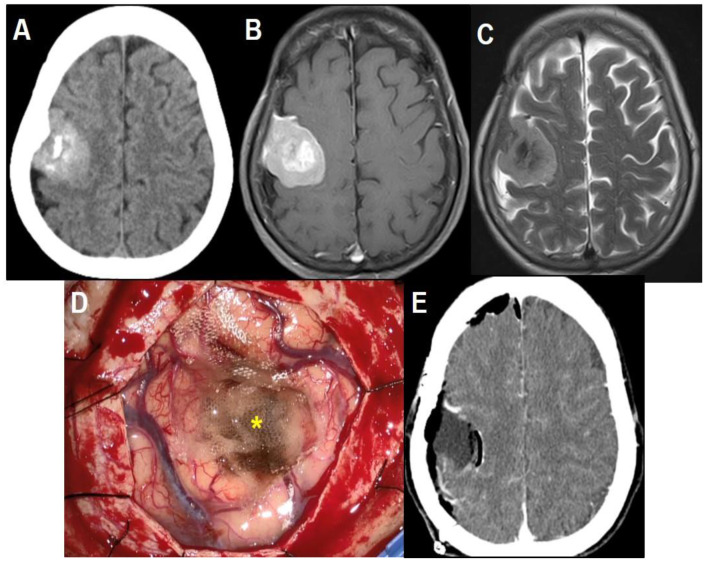
Perioperative images. (**A**) Brain CT scan, and (**B**) gadolinium-enhanced T1-weighted and (**C**) T2-weighted axial MR images showing a homogeneous enhanced, calcified, and multi-lobulated mass adjacent to the right motor strip without perilesional edema. (**D**) Intraoperative photograph demonstrating gross total resection without definitive cortical injury. Small arachnoid disruption and venous bleeding was controlled by a fibrin sealant patch and oxidized cellulose at the mass stump. (**E**) Brain CT scan immediately after resection showing air-bubbles in the resection bed, which were due to hemostatic agents. This is indicated by an asterisk in the operative photograph.

**Figure 2 brainsci-11-00299-f002:**
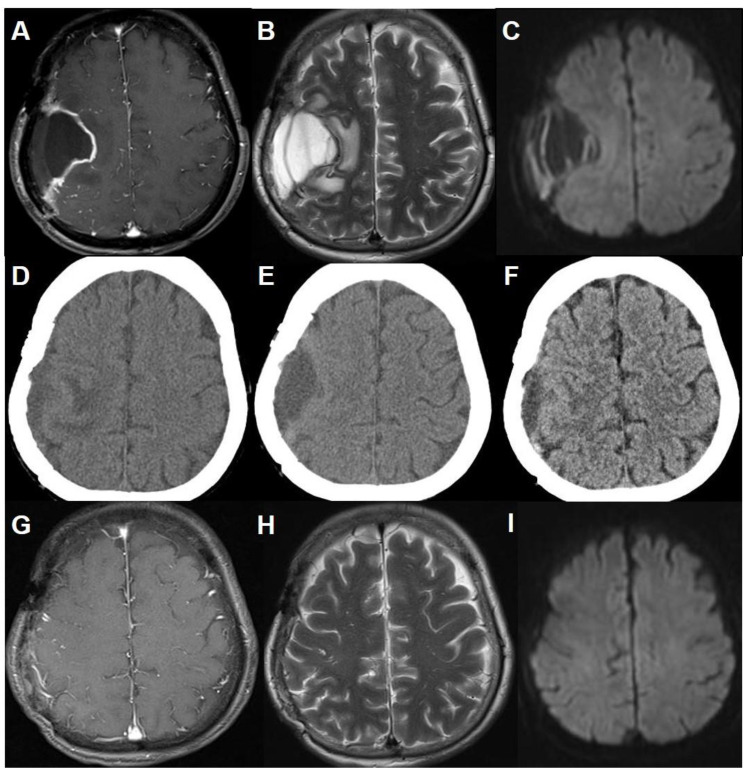
Time course of late-onset, localized subdural fluid collection in the resection cavity. (**A**) Gadolinium-enhanced T1-weighted and (**B**) T2-weighted axial MR images showing a subdural fluid collection at the resection cavity with an enhanced cavity margin and perilesional edema. (**C**) There was no diffusion restriction on the diffusion-weighted image. Brain CT scans taken after two weeks (**D**), four weeks (**E**), and three months (**F**) of conservative management show the spontaneous regression of the fluid collection and perilesional edema. Note that s temporary increase of the fluid collection was noted at discontinuing conservative therapy which was resolved after retreatment. (**G–I**). The latest MR images demonstrating the complete disappearance of the fluid collection without perilesional edema and recurrence.

**Figure 3 brainsci-11-00299-f003:**
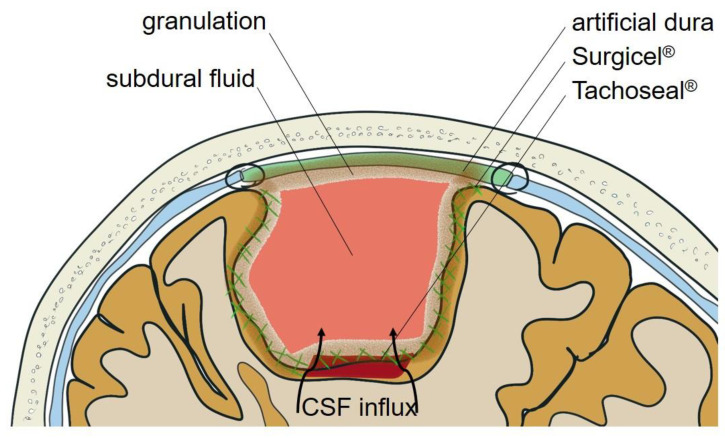
Schematic illustration for possible pathogenesis. Cerebrospinal fluid (CSF) was flowed in through the tearing of the arachnoid and accumulated at the resection cavity. The efflux of CSF into the normal arachnoid space was interrupted by the reactive granulomatous inflammation, induced by hemostatic agents and artificial dura.

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
