# Peer review of "Spontaneous Resolution of Late-Onset, Symptomatic Fluid Collection Localized in the Meningioma Resection Cavity: A Case Report and Suggestion of Possible Pathogenesis"

_brainsci, 2021, doi:10.3390/brainsci11030299_

Round 1

Reviewer 1 Report

All the various mechanisms suspected to be involved in the onset the fluid collection localized at the resection site after convexity meningioma surgery are well described .

But I completely disagree with  the management of the symptomatic fluid collection: to solve the problem the patient underwent for two weeks mannitol and steroid and for another two months prednisolone 10 mg per day.

It is a too long time of a  heavy therapy, when you could solve the problem with a very simply and safe surgical procedure.

Author Response

Thank you for kind comments. We did not define definitive treatment guideline for symptomatic fluid collection in the resection cavity after meningioma surgery, but just suggest conservative treatment as a feasible treatment options. As reviewer’s comment, surgical treatment was tried to symptomatic necrotizing lesions due to the granulomatous inflammation (new ref. 7). However, serial radiologic MR findings demonstrated that this granulomatous inflammation could resolved with time (new ref. 8). We think that conservative management can be considered as an initial treatment strategy for this complication. We add the sentences (page 5, line 146-151) with new references in discussion section to clarify this point. 

Reviewer 2 Report

In this case report the authors discussed a symptomatic fluid collection after brain meningioma resection. The patient was conservatively managed with progressive resolution of the fluid collection after steroid therapy.

During surgery, the surgeons placed fibrin sealant and oxidize cellulose in the resection cavity and a dural substitute.

The topic is potentially interesting and the case weel documented.

The discussion provides a series of alternative mechanisms that can be at the basis of fluid collection in the resection cavity, however in the case discussed by the authors the only reason is the placement of multiple heterologous materials, which usually cause an inflammatory reaction and mass effect. With steroids administration, the authors probably reduce the immune reaction and for this reasons the patient did well and the collection resolved.

The authors should discuss the role of medical management and, due to the rarity of the case, which is not new in the pertinent literature, they should discuss the previous cases published and how they have been managed.

After this revision, in my opinion the article can reach enough quality.

Author Response

Thank you for considerate comments. We suggested that arachnoid tear with flap valve formation, dura-arachnoid interface, and granulomatous inflammation may induce localized subdural fluid collection. In our case, arachnoid membrane might be injured at bleeding site with flap valve formation. Dura-arachnoid interface was formed in resection cavity. And granulomatous inflammation was reinforced by blood components and exogenous materials. To clarify intent, we add the sentence (page 4, line 124) in discussion section.

To the best of our knowledge, there were no definitive treatment strategy for this com-plication based on the pertinent literature. In some cases, surgical treatment was tried to symptomatic necrotizing lesions due to the granulomatous inflammation (new ref. 7). However, serial radiologic MR findings demonstrated that this granulomatous inflammation could resolved with time (new ref. 8). To clarify this point, we added new sentences in discussion section (page 5, line 146-151).

Round 2

Reviewer 1 Report

After the revision, the article  reachs  enough quality.

Reviewer 2 Report

The authors improved the article